# A Coach-Player Framework for Dynamic Team Composition

## Abstract

In real-world multi-agent teams, agents with different capabilities may join or leave "on the fly" without altering the team's overarching goals. Coordinating teams with such *dynamic composition* remains a challenging problem: the optimal team strategy may vary with its composition. Inspired by real-world team sports, we propose a coach-player framework to tackle this problem. We assume that the players only have a partial view of the environment, while the coach has a complete view. The coach coordinates the players by distributing individual *strategies*. Specifically, we 1) propose an attention mechanism for both the players and the coach; 2) incorporate a variational objective to regularize learning; and 3) design an adaptive communication method to let the coach decide when to communicate with different players. Our attention mechanism on the players and the coach allows for a varying number of heterogeneous agents, and can thus tackle the dynamic team composition. We validate our methods on resource collection tasks in multi-agent particle environment. We demonstrate zero-shot generalization to new team compositions with varying numbers of heterogeneous agents. The performance of our method is comparable or even better than the setting where all players have a full view of the environment, but no coach. Moreover, we see that the performance stays nearly the same even when the coach communicates as little as 13% of the time using our adaptive communication strategy. These results demonstrate the significance of a coach to coordinate players in dynamic teams.

## 1 Introduction

Cooperative multi-agent reinforcement learning (MARL) is the problem of coordinating a team of agents to perform a shared task. It has broad applications in autonomous vehicle teams (Cao et al., 2012), sensor networks (Choi et al., 2009), finance (Lee et al., 2007), and social science (Leibo et al., 2017). Recent works in multi-agent reinforcement learning (MARL) have shed light on solving challenging multi-agent problems such as playing StarCraft with deep learning models (Rashid et al., 2018). Among these methods, centralized training with decentralized execution (CTDE) has gained much attention since learning in a centralized way enables better cooperation while executing independently makes the system efficient and scalable (Lowe et al., 2017). However, most deep CTDE approaches for cooperative MARL are limited to a fixed number of homogeneous agents.

Real-world multi-agent tasks, on the other hand, often involve dynamic teams. For example, in a soccer game, a team receiving a red card has one fewer player. In this case, the team may switch to a more defensive strategy. As another example, consider an autonomous vehicle team for delivery. The control over the team depends on how many vehicles we have, how much load each vehicle permits, as well as the delivery destinations. In both examples, the optimal team strategy varies according to the *team composition*,[1] i.e., the size of the team and each agent's capability. In these settings, it is intractable to re-train the agents for each new team composition, and it is thus desirable to have zero-shot generalization to new team compositions that are not seen during training.

Recently, Iqbal et al. (2020) proposed a multi-head attention model for learning in environments with a variable number of agents under the CTDE framework. However, in many challenging tasks,

---

[1]Team composition is part of an environmental scenario (de Witt et al., 2019), which also includes other environment entities. The formal definition is in Section 2.1.

Figure 1: (a) In training, we sample teams from a set of compositions. The coach observes the entire world and coordinates different teams via broadcasting strategies periodically; (b) A team with dynamic composition can be viewed as a sequence of fixed composition team, thus the proposed training generalizes to dynamic composition; (c) Our method is at the star position within MARL.[3]

the CTDE constraint is too restrictive as each agent only has access to its own decisions and partial environmental observations at test time – See Section 3.1 for an example where this requirement causes failure to learn. The CTDE constraint can be relaxed either by 1) allowing all agents to communicate with each other (Zhang et al., 2018) or 2) having a special "coach" agent who distributes strategic information based on the full view of the environment (Stone & Veloso, 1999). The former case is typically too expensive for many CTDE scenarios (e.g., battery-powered drones or vehicles), while the latter case of having a coach may be feasible (e.g., satellites or watchtowers to monitor the field in which agents operate). In this work, we focus on the latter approach of having a coach to coordinate the agents.

Specifically, we grant the coach with *global* observation while agents only have partial views of the environment. We assume that the coach can distribute information to various agents only in limited amounts. We model this communication through a continuous vector, termed as the strategy vector, and it is specific to each agent. We design each agent's decision module to incorporate the most recent strategy vector from the coach. We further propose a variational objective to regularize learning, inspired by (Rakelly et al., 2019; Wang et al., 2020a). In order to save costs incurred in receiving information from the coach, we additionally design an adaptive policy where the coach communicates with different players only as needed. To train the coach and agents, we sample different teams from a set of team compositions. Recall that the training is centralized under the CTDE framework. At execution time, the learned policy generalizes across different team compositions in a zero-shot manner. Our framework also allows for dynamic teams whose composition varies over time (see Figure 1 (a-b)).

**Summary of Results:** We (1) propose a coach-player framework for dynamic team composition of heterogeneous agents; (2) introduce a variational objective to regularize the learning, which leads to improved performance; (3) design an adaptive communication strategy to minimize communication from the coach to the agents. We apply our methods on resource-collection tasks in multi-agent particle environments. We evaluate zero-shot generalization for new team compositions at test time. Results show comparable or even better performance against methods where players have full observation but no coach. Moreover, there is almost no performance degradation even when the coach communicates as little as 13% of the time with the players. These results demonstrate the effectiveness of having a coach in dynamic teams.

## 2 BACKGROUND

### 2.1 PROBLEM FORMULATION

We model the cooperative multi-agent task under the *Decentralized partially observable Markov Decision Process* (Dec-POMDP) (Oliehoek et al., 2016). Specifically, we build on the setting of Dec-POMDP with entities (de Witt et al., 2019), which considers entity-based knowledge representation. Here, entities include both controllable agents and other environment landmarks. In addition, we extend the representation to allow agents to have individual characteristics, i.e., skill-level, physical condition, etc. Therefore, a Dec-POMDP with characteristics-based entities can be described as a tuple $(\boldsymbol{S}, \boldsymbol{U}, \boldsymbol{O}, P, R, \mathcal{E}, \mathcal{A}, \mathcal{C}, m, \Omega, \rho, \gamma)$. $\mathcal{E}$ represents the space of entities. $\forall e \in \mathcal{E}$, the entity $e$ has its state representation $s^e \in \mathbb{R}^{d_e}$. The global state is therefore the set $\boldsymbol{s} = \{s^e | e \in \mathcal{E}\} \in \boldsymbol{S}$.

---

[3]Rigorously speaking the players in our method occasionally receive global information from the coach. But players still execute independently with local views while they benefit from the centralized learning.

A subset of the entities are controllable agents $a \in \mathcal{A} \subseteq \mathcal{E}$. For both agents and non-agent entities, we differentiate them based on their characteristics $c^e \in \mathcal{C}$.[4] For example, $c^e$ can be a continuous vector that consists of two parts such that only one part can be non-zero vector. That is, if $e$ is an agent, the first part can represent its skill-level or physical condition, and if $e$ is an non-agent entity, the second part can represent its entity type. A scenario is a multiset of entities $\boldsymbol{c} = \{c^e | e \in \mathcal{E}\} \in \Omega$ and possible scenarios are drawn from the distribution $\rho(\boldsymbol{c})$. In other words, scenarios are unique up to the composition of the team and that of world entities. Fixing any particular scenario $\boldsymbol{c}$, it maps to a normal Dec-POMDP with the fixed multiset of entities $\{e | c^e \in \boldsymbol{c}\}$.

Given a scenario $\boldsymbol{c}$, at each environment step, each agent $a$ can observe a subset of entities specified by an observability function $m : \mathcal{A} \times \mathcal{E} \to \{0, 1\}$, where $m(a, e)$ indicates whether agent $a$ can observe entity $e$.[5] Therefore, an agent's observation is a set $o^a = \{s^e | m(a, e) = 1\} \in \boldsymbol{O}$. All agents can perform the joint action $\boldsymbol{u} = \{u^a | a \in \mathcal{A}\} \in \boldsymbol{U}$, and the environment will step according to the transition dynamics $P(\boldsymbol{s}'|\boldsymbol{s}, \boldsymbol{u}; \boldsymbol{c})$. After that, the entire team will receive a single scalar reward $r \sim R(\boldsymbol{s}, \boldsymbol{u}; \boldsymbol{c})$. Starting from an initial state $\boldsymbol{s}_0$, the MARL objective is to maximize the discounted cumulative team reward over time: $G = \mathbb{E}_{\boldsymbol{s}_0, \boldsymbol{u}_0, \boldsymbol{s}_1, \boldsymbol{u}_1, \dots}[\sum_{t=0}^{\infty} \gamma^t r_t]$, where $\gamma$ is the discount factor. Our goal is to learn a team policy that can generalize across different scenarios $\boldsymbol{c}$ (different team compositions) and eventually dynamic scenarios (varying team compositions over time).

For optimizing $G$, $Q$-learning is a specific method that learns an accurate action-value function and makes decision based on that. The optimal action-value function $Q$ satisfies the Bellman equality: $Q_*^{\text{tot}}(\boldsymbol{s}, \boldsymbol{u}; \boldsymbol{c}) = r(\boldsymbol{s}, \boldsymbol{u}; \boldsymbol{c}) + \gamma \mathbb{E}_{\boldsymbol{s}' \sim P(\cdot|\boldsymbol{s}, \boldsymbol{u}; \boldsymbol{c})} [\max_{\boldsymbol{u}'} Q_*^{\text{tot}}(\boldsymbol{s}', \boldsymbol{u}'; \boldsymbol{c})]$, where $Q_*^{\text{tot}}$ denote the team's optimal $Q$-value. A common strategy is to adopt function approximation and parameterize the optimal $Q_*^{\text{tot}}$ with parameter $\theta$. Moreover, due to partial observability, the history of observation-action pairs is often encoded to a compact vector representation, i.e., via a recurrent neural network (Medsker & Jain, 1999), in place of the state: $Q_\theta^{\text{tot}}(\boldsymbol{\tau}_t, \boldsymbol{u}_t; \boldsymbol{c}) \approx \mathbb{E}[Q_*^{\text{tot}}(\boldsymbol{s}_t, \boldsymbol{u}_t; \boldsymbol{c})]$, where $\boldsymbol{\tau} = \{\tau^a | a \in \mathcal{A}\}$ and $\tau^a = (o_0^a, u_0^a, \dots o_t^a)$. In practice, at each time step $t$, the recurrent neural network takes in $(u_{t-1}^a, o_t^a)$ as the new input, where $u_{-1}^a = \boldsymbol{0}$ at $t = 0$ (Zhu et al., 2017). Deep $Q$-learning (Mnih et al., 2015) uses deep neural networks to approximate the $Q$ function, its objective in our case is:

$$\mathcal{L}(\theta) = \mathbb{E}_{(\boldsymbol{c}, \boldsymbol{\tau}_t, \boldsymbol{u}_t, r_t, \boldsymbol{\tau}_{t+1}) \sim \mathcal{D}}\left[\left(r_t + \gamma \max_{\boldsymbol{u}'} Q_{\bar{\theta}}^{\text{tot}}(\boldsymbol{\tau}_{t+1}, \boldsymbol{u}'; \boldsymbol{c}) - Q_\theta^{\text{tot}}(\boldsymbol{\tau}_t, \boldsymbol{u}_t; \boldsymbol{c})\right)^2\right]. \quad (1)$$

Here, $\mathcal{D}$ is a replay buffer that stores previously generated off-policy data. $Q_{\bar{\theta}}^{\text{tot}}$ is the target network parameterized by a delayed copy of $\theta$ for stability.

## 2.2 VALUE FUNCTION FACTORIZATION AND ATTENTION QMIX

Factorizing the action-value function $Q$ into per agent value function has become a popular approach in centralized training and decentralized execution. Specifically, Rashid et al. (2018) proposes QMIX that factorizes $Q^{\text{tot}}(\boldsymbol{\tau}_t, \boldsymbol{u})$ into $\{Q^a(\tau_t^a, u^a | a \in \mathcal{A}\}$ and combines them via a mixing network such that $\forall a, \frac{\partial Q_{\text{tot}}}{\partial Q^a} \geq 0$. The condition guarantees that individual optimal action $u^a$ is also the best action for the team. As a result, during execution, the mixing network can be removed and agents work independently according to their own $Q^a$. Attention QMIX (A-QMIX) (Iqbal et al., 2020) augments the QMIX algorithm with attention mechanism to deal with an indefinite number of agents/entities. In particular, for each agent, the algorithm applies the multi-head attention (MHA) layer (Vaswani et al., 2017) to summarize the information of the other entities. This information is used for both encoding the agent's state and adjusting the mixing network. Specifically, the input $\boldsymbol{o}$ is represented by two matrices: the entity state matrix $\boldsymbol{X}^{\mathcal{E}}$ and the observability matrix $\boldsymbol{M}$. Assume at the given scenario $\boldsymbol{c}$, there exists $n_e$ entities, $n_a$ of which are the controllable agents, then $\boldsymbol{X}^{\mathcal{E}} \in \mathbb{R}^{n_e \times d_e}$ includes all entities encoding and the first $n_a$ rows belong to agents. $\boldsymbol{M} \in \{0, 1\}^{n_a \times n_e}$ is a binary observability mask and $M_{ij} = m(a^i, e^j)$ indicates whether agent $i$ observes entity $j$. $\boldsymbol{X}^{\mathcal{E}}$ is first passed through an encoder, i.e., a single-layer feed-forward network, and becomes $\boldsymbol{X}$. Denote the $k$-th row of $\boldsymbol{X}$ as $h_k$, then for the $i$-th agent, the MHA layer then takes $h_i$ as the query and $\{h_j | M_{ij} = 1\}$ as the keys to compute a latent representation of $a^i$'s observation. For the mixing network, the same MHA layer will take $\boldsymbol{X}^{\mathcal{E}}$ and the full observation

---

[4]$c^e$ is part of $s^e$, but we will explicitly write out $c^e$ in the following for emphasis.
[5]An agent can always observe itself, i.e., $m(a, a) = 1, \forall a \in \mathcal{A}$.

matrix $\boldsymbol{M}^*$, where $\boldsymbol{M}_{ij}^* = 1$ if both $e^i$ and $e^j$ exist in the scenario $\boldsymbol{c}$, and outputs the encoded global representation for each agent. These encoded representations are then used to generate the mixing network. We refer readers to Appendix B of Iqbal et al. (2020) for more details. While A-QMIX in principle applies to the dynamic team composition problem, it is restricted to fully decentralized execution with partial observation. We borrow the attention modules from A-QMIX but additionally investigate how to efficiently take advantage of the global information by introducing the coach.

Iqbal et al. (2020) proposes an extended version of A-QMIX, called Attentive-Imaginative QMIX (AI-QMIX), which randomly breaks up the team into two disjoint parts for each agent's $Q^a$ to further decompose the $Q$ value. While the authors demonstrate AI-QMIX outperforms A-QMIX on a gridworld resource allocation task and a modified StarCraft environment. As we will show in the experiment section, we find that AI-QMIX does not improve over A-QMIX by much while doubling the computation resource. For this reason, our method is mainly based on the A-QMIX framework, but extending it to AI-QMIX is straightforward.

## 3 METHOD

Here we present the coach-player architecture to incorporate global information for adapting the team-level strategy across different scenarios $\boldsymbol{c}$. We first introduce the coach agent that coordinates base agents with global information via broadcasting strategies periodically. Then we present the learning objective and and an additional variational objective to regularize the training. We finish by introducing a method to reduce the broadcast rate and provide analysis to support it.

### 3.1 ON THE IMPORTANCE OF GLOBAL INFORMATION

As the optimal team strategy varies according to the scenario $\boldsymbol{c}$, which includes the team composition, it is important for the team to be aware of the scenario change promptly. In an extreme example, assume in a multi-agent problem where every agent has its skill-level represented by a real number $c^a \in \mathbb{R}$ and there is a task to complete. For each agent $a$, $u^a \in \{0, 1\}$ indicates whether $a$ chooses to perform the task. The reward is defined as $R(\boldsymbol{u}; \boldsymbol{c}) = \max_a c^a \cdot u^a + 1 - \sum_a u^a$. In other words, the reward is proportional to the skill-level of the agent who performs it and the team got penalized if more than 1 agent choose to perform the task. If the underlying scenario $\boldsymbol{c}$ is fixed, even if all agents are unaware of others' capabilities, it is still possible for the team to gradually figure out the optimal strategy. By contrast, when $\boldsymbol{c}$ is subject to change, i.e. agents with different $c$ can join or leave, even if we allow agents to communicate via a network, the information that a particular agent joins or leaves generally takes $d$ time steps to propagate where $d$ is the longest shortest path from that agent to any other agents. Therefore, we can see that knowing the global information is not only beneficial but sometimes also necessary for coordination. This motivates the introduction of the coach agent.

### 3.2 COACH AND PLAYERS

We introduce a coach agent and grant it with global observation. To preserve efficiency as in the decentralized setting, we limit the coach agent to only distribute information via a continuous vector $z^a \in \mathbb{R}^{d_z}$ ($d_z$ is the dimension of strategy) to agent $a$, which we call the strategy, once every $T$ time steps. $T$ is the communication interval. The team strategy is therefore represented as $\boldsymbol{z} = \{z^a | a \in \mathcal{A}\}$. Strategies are predicted via a function $f$ parameterized by $\phi$. Specifically, we assume

$$z^a \sim \mathcal{N}(\mu^a, \Sigma^a), \quad \text{where} \quad (\boldsymbol{\mu} = \{\mu^a | a \in \mathcal{A}\}, \boldsymbol{\Sigma} = \{\Sigma_a | a \in \mathcal{A}\}) = f_\phi(\boldsymbol{s}; \boldsymbol{c}). \quad (2)$$

Within the next $T$ steps, agent $a$ will act conditioned on the strategy $z^a$. Specifically, within an episode, at time $t_k \in \{v | v \equiv 0 \pmod{T}\}$, the coach observes the global state $\boldsymbol{s}_{t_k}$ and computes and distributes the strategies $\boldsymbol{z}_{t_k}$ for all agents. From time $t \in [t_k, t_k + T - 1]$, any agent $a$ will act according to its individual action-value $Q^a(\tau_t^a, \cdot \mid z_{t_k}^a; c^a)$.

Denote $\hat{t} = \max\{v | v \equiv 0 \pmod{T} \text{ and } v \leq t\}$, the most recent time step when the coach distribute strategies. The mean square Bellman error objective in equation 1 becomes

$$\mathcal{L}_{\text{RL}}(\theta, \phi) = \mathbb{E}_{(\boldsymbol{c}, \boldsymbol{\tau}_t, \boldsymbol{u}_t, r_t, \boldsymbol{s}_{\hat{t}}, \boldsymbol{s}_{\hat{t}+1}) \sim \mathcal{D}} \left[ \left( r_t + \gamma \max_{\boldsymbol{u}'} Q_\theta^{\text{tot}}(\boldsymbol{\tau}_{t+1}, \boldsymbol{u}' | \boldsymbol{z}_{\widehat{t+1}}; \boldsymbol{c}) - Q_\theta^{\text{tot}}(\boldsymbol{\tau}_t, \boldsymbol{u}_t \mid \boldsymbol{z}_{\hat{t}}; \boldsymbol{c}) \right)^2 \right],$$

$$(3)$$

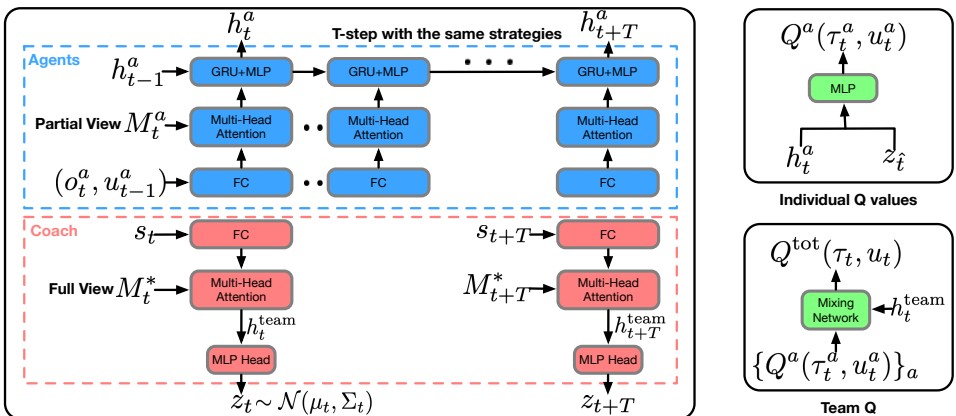

Figure 2: The coach-player network architecture. Here, GRU refers to gated recurrent unit (Chung et al., 2014); MLP refers to multi-layer perceptron; FC refers to fully connected layer. Both coach and players use multi-head attention to encode information. The coach has full view while players have partial views. $h_t^a$ encodes agent $a$'s history. $h_t^a$ combines the most recent strategy $z_{\hat{t}} = z_{t-t\%T}$ to predict the individual utility $Q^a$. The mixing network combines all $Q^a$s to predict $Q^{\text{tot}}$.

where $z_{\hat{t}} \sim f_\phi(s_{\hat{t}}; c)$, $z_{t\hat{+}1} \sim f_{\bar{\phi}}(s_{t\hat{+}1}; c)$, and $\bar{\phi}$ is the parameter of the target network for the coach's strategy predictor $f$. We build our network on top of A-QMIX but use a separate multi-head attention (MHA) layer to encode the global states that the coach observes. For the mixing network, we also use the coach's output from the MHA layer for mixing the individual $Q^a$ to form the team $Q^{\text{tot}}$. The entire architecture is described in Figure 3.2. We provide more details in Appendix.

## 3.3 REGULARIZING WITH VARIATIONAL OBJECTIVE

Inspired by recent work that applied variational inference to regularize the learning of a latent space in reinforcement learning (Rakelly et al., 2019; Wang et al., 2020a), we also introduce a variational objective to stabilize the training. Intuitively, an agent's behavior should be consistent with its assigned strategy. In other words, the received strategy should be identifiable from the agent's future trajectory. Therefore, we propose to maximize the mutual information between the strategy and the agent's future observation-action pairs $\zeta_t^a = (o_{t+1}^a, u_{t+1}^a, o_{t+2}^a, u_{t+2}^a, \ldots, o_{t+T-1}^a, u_{t+T-1}^a)$. We maximize the following variational lower bound:

$$
\begin{aligned}
I(z_t^a; \zeta_t^a, s_t) &= \mathbb{E}_{s_t, z_t^a, \zeta_t^a} \left[ \log \frac{q_\xi(z_t^a | \zeta_t^a, s_t)}{p(z^a | s_t)} \right] + D_{\text{KL}}\left( p(z_t^a | \zeta_t^a, s_t), q_\xi(z_t^a | \zeta_t^a, s_t)) \right) \\
&\geq \mathbb{E}_{s_t, z_t^a, \zeta_t^a} \left[ \log \frac{q_\xi(z_t^a | \zeta_t^a, s_t)}{p(z^a | s_t)} \right] = \mathbb{E}_{s_t, z_t^a, \zeta_t^a} \left[ \log q_\xi(z_t^a | \zeta_t^a, s_t) \right] + H(z_t^a | s_t).
\end{aligned}
\tag{4}
$$

Here $H(\cdot)$ denotes the entropy and $q_\xi$ is the variational distribution parameterized by $\xi$. We further adopt the Gaussian factorization for $q_\xi$ as in (Rakelly et al., 2019), i.e. $q_\xi(z_t^a | \zeta_t^a, s_t) \propto q_\xi^{(t)}(z_t^a | s_t, u_t^a) \prod_{k=t+1}^{t+T-1} q_\xi^{(k)}(z_t^a | o_k^a, u_k^a)$, where each $q_\xi^{(\cdot)}$ is a Gaussian distribution. So $q_\xi$ predicts the $\hat{\mu}_t^a$ and $\hat{\Sigma}_t^a$ of a multivariate normal distribution from which we calculate the log-probability of $z_t^a$. In practice, $z_t^a$ is sampled from $f_\phi$ using the re-parameterization trick (Kingma & Welling, 2013). The objective is $\mathcal{L}_{\text{var}}(\phi, \xi) = -\lambda_1 \mathbb{E}_{s_t, z_t^a, \zeta_t^a} [\log q_\xi(z_t^a | \zeta_t^a, s_t)] - \lambda_2 H(z_t^a | s_t)$, where $\lambda_1$ and $\lambda_2$ are tunable coefficients.

## 3.4 REDUCING THE COMMUNICATION FREQUENCY

So far, we assume at every $T$ steps the coach periodically broadcasts new strategies for all agents. In practice, broadcasting suffers communication cost or bandwidth limit. So it is desirable to only distribute strategies when "necessary". To reduce the communication frequency, we propose an intuitive method that decides whether to distribute new strategies based on the $\ell_2$ distance of the old strategy to the new one. In particular, at time step $t = kT, k \in \mathbb{Z}$, assuming the prior strategy for agent $a$ is $z_{\text{old}}^a$, the new strategy for agent $a$ is

$$
\tilde{z}_t^a = \begin{cases} z_t^a \sim f_\phi(s, c) & \text{if } ||z_t^a - z_{\text{old}}^a||_2 \leq \beta \\ z_{\text{old}}^a & \text{otherwise.} \end{cases}
\tag{5}
$$

For a general time step $t$, the individual strategy for $a$ is therefore $\tilde{z}_{\hat{t}}^a$. Here $\beta$ is a manually specified threshold. Note that we can train a single model and apply this criterion for all agents. By adjusting $\beta$, one can easily achieve different communication frequencies. Intuitively, when the previous strategy is "close" to the current one, it should be more tolerant to keep using it. The intuition is concrete when the learned $Q_{\theta}^{\text{tot}}$ has relatively small Lipschitz constant. If we assume $\forall \boldsymbol{\tau}_t, \boldsymbol{u}_t, \boldsymbol{s}_t, \boldsymbol{s}_{\hat{t}}, \boldsymbol{c}, ||Q^{\text{tot}}(\boldsymbol{\tau}_t, \boldsymbol{u}_t, f(\boldsymbol{s}_{\hat{t}}); \boldsymbol{c}) - Q_*^{\text{tot}}(\boldsymbol{s}_t, \boldsymbol{u}_t; \boldsymbol{c})||_2 \leq \kappa$, where $Q_*^{\text{tot}}$ is the optimal $Q$, and $\forall z_1^a, z_2^a, |Q^{\text{tot}}(\boldsymbol{\tau}_t, \boldsymbol{u}_t | z_1^a, \boldsymbol{z}^{-a}; \boldsymbol{c}) - Q^{\text{tot}}(\boldsymbol{\tau}_t, \boldsymbol{u}_t | z_2^a, \boldsymbol{z}^{-a}; \boldsymbol{c})| \leq \eta ||z_1^a - z_2^a||_2$, we have the following:

**Theorem 1.** *If the used team strategies $\tilde{\boldsymbol{z}}_t$ satisfies $\forall a, t, ||\tilde{z}_{\hat{t}}^a - z_{\hat{t}}^a||_2 \leq \beta$, denote the action-value and the value function of following the used strategies as $\tilde{Q}$ and $\tilde{V}$, i.e. $\tilde{V}(\boldsymbol{\tau}_t | \tilde{\boldsymbol{z}}_{\hat{t}}; \boldsymbol{c}) = \max_{\boldsymbol{u}} \tilde{Q}(\boldsymbol{\tau}_{\hat{t}}, \boldsymbol{u} | \tilde{\boldsymbol{z}}_t; \boldsymbol{c})$, and define $V_*^{tot}$ similarly, we have*

$$||V_*^{tot}(\boldsymbol{s}_t; \boldsymbol{c}) - \tilde{V}(\boldsymbol{\tau}_t | \tilde{\boldsymbol{z}}_{\hat{t}}; \boldsymbol{c})||_\infty \leq \frac{2(n_a \eta \beta + \kappa)}{1 - \gamma}, \tag{6}$$

*where $n_a$ is the number of agents and $\gamma$ is the discount factor.*

We defer the proof to Appendix A. The method described in equation 5 satisfies the condition in Theorem 1 and therefore when $\beta$ is small, distributing strategies according to equation 5 will not result in much performance drop.

## 4 EXPERIMENTS

We design the experiments to 1) verify the effectiveness of the coach agent; 2) investigate how performance varies with the interval $T$; 3) test if the variational objective is useful; and to 4) understand how much the performance drops by adopting the method in equation 5. We test our idea on a resource collection task with different scenarios in customized multi-agent particle environments (Lowe et al., 2017). In the following, we call our method COPA (COach-and-PlAyer).

### 4.1 RESOURCE COLLECTION

In Resource Collection, a team of agents coordinate to collect different resources spread out on a square map with width $1.8$. There are 4 types of entites: the resources, the agents, the home and the invader. We assume there are 3 types of resources: $(r)$ed, $(g)$reen and $(b)$lue. In the world, always 6 resources appear with 2 of each type. Each agent has 4 characteristics $(c_r^a, c_g^a, c_b^a, v^a)$, where $c_x^a$ represents how efficient $a$ collects the resource $x$ and $v$ is the agent's *max* moving speed. The agent's job is to collect the most amount of resources and bring them home, and catch the invader if it appears. If $a$ collects $x$, the team receives a reward of $10 \cdot c_x^a$ as reward. Holding any resource, agents cannot collect more and need to bring the resource home until going out again. Bringing a resource home has $1$ reward. Occasionally the invader appears and goes directly to home. Any agent catch the invader will have $4$ reward. If the invader reaches home, the team is penalized by $-4$ reward. Each agent has 5 actions: accelerate up / down / left / right and decelerate, and it observes anything within $0.2$ distance. The maximum episode length is $145$. In training, we allow scenarios to have 2 to 4 agents, and for each agent, $c_r^a, c_g^a, c_b^a$ are chosen from $\{0.1, 0.5, 0.9\}$ and the max speed $v^a$ from $\{0.3, 0.5, 0.7\}$. We design 3 testing tasks: 5-agent task, 6-agent task, and a varying-agent task. For each task, we generate 1000 different scenarios $\boldsymbol{c}$. Each scenario includes $n_a$ agents, 6 resources and an invader. For agents, $c_r^a, c_g^a, c_b^a$ are chosen uniformly from the interval $[0.1, 0.9]$ and $v^a$ from $[0.2, 0.8]$. For a particular scenario in the varying agent task, starting from 4 agents, the environment randomly adds or drops an agent every $\nu$ steps as long as the number of agents remains in $[2, 6]$. $\nu$ is a random variable from the uniform distribution $\mathcal{U}(8, 12)$. See Figure 3 for an example run of the learned policy.

**Effectiveness of Coach** We provide the training curve in Figure 4.1 (a) where the communication interval is set to $T = 4$. The black solid line is a hard-coded greedy algorithm where agents always go for the resource they are mostly good at collecting, and whenever the invader appears, the closest agent goes for it. We see that without global information, A-QMIX and AI-QMIX are significantly below the hard-coded baseline. Without the coach, we let all agents have the global view every $T$ steps in A-QMIX (periodic) but it barely improves over A-QMIX. A-QMIX (full) is fully centralized, i.e., all agents have global view. Without the variational objective, COPA is comparable against A-QMIX (full). With the variational objective, it becomes even better than

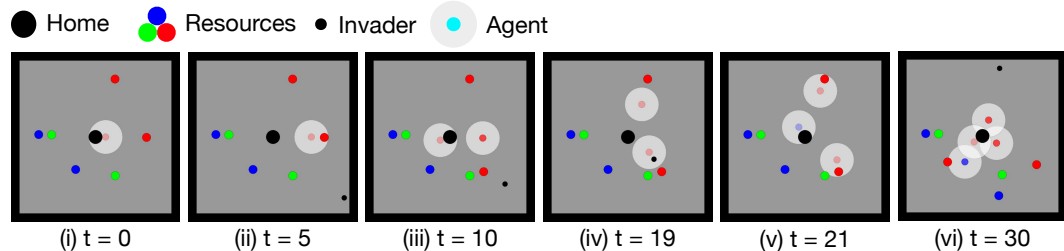

Figure 3: An example episode up to $t = 30$ with communication interval $T = 4$. Here, $c^a$ is represented by rgb values, $c^a = (r, g, b, v)$. For illustration, we set agents rgb to be one-hot but it can vary in practice. (i) an agent starts at home; (ii) the invader (black) appears while the agent (red) goes to the red resource; (iii) another agent is spawned while the old agent brings resource home; (iv) one agent goes for the invader while the other for resource; (v-vi) a new agent (blue) is spawned and goes for the blue resource while other agents (red) are bringing resources home.

A-QMIX (full). Note that all baseline methods are scaled to have more parameters than COPA. The results demonstrate the importance of global coordination and the coach-player hierarchy.

**Communication Interval** To investigate how performance varies with $T$, we train with different $T$ chosen from $[2, 4, 8, 12, 16, 20, 24]$ in Figure 4.1(b). Interestingly, the performance peaks at $T = 4$, contradicting the intuition that smaller $T$ is better. This shows the coach is more useful when it can make the agents behavior smooth/consistent over time.

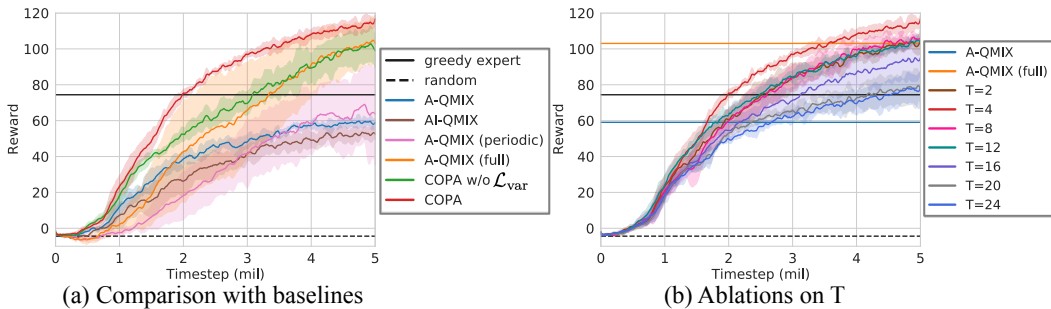

(a) Comparison with baselines        (b) Ablations on T

Figure 4: Training curves for Resource Collection. (a) comparison against A-QMIX, AI-QMIX and COPA without the variational objective. Here we choose $T = 4$; (b) ablations on the communication interval $T$. All results are averaged over 5 seeds.

| Method | Env. ($n = 5$) | | Env. ($n = 6$) | | Env. (varying $n$) | |
|---|---|---|---|---|---|---|
| | Reward | Comm. Frequency | Reward | Comm. Frequency | Reward | Comm. Frequency |
| Random Policy | 6.9 | N/A | 10.4 | N/A | 2.3 | N/A |
| Greedy Expert | 115.3 | N/A | 142.4 | N/A | 71.6 | N/A |
| AI-QMIX | 90.5±1.5 | 0. | 109.3±1.6 | 0. | 61.5±0.9 | 0. |
| A-QMIX | 96.9±2.1 | 0. | 115.1±2.1 | 0. | 66.2±1.6 | 0. |
| A-QMIX (periodic) | 93.1±20.4 | 0.25 | 104.2±22.6 | 0.25 | 68.9±12.6 | 0.25 |
| A-QMIX (full) | 157.4±8.5 | 1. | 179.6±9.8 | 1. | 114.3±6.2 | 1. |
| COPA ($\beta = 0$) | 175.6±1.9 | 0.25 | 203.2±2.5 | 0.25 | 124.9±0.9 | 0.25 |
| COPA ($\beta = 2$) | 174.4±1.7 | 0.18 | 200.3±1.6 | 0.18 | 122.8±1.5 | 0.18 |
| COPA ($\beta = 3$) | 168.8±1.7 | 0.13 | 195.4±1.8 | 0.13 | 120.0±1.6 | 0.14 |
| COPA ($\beta = 5$) | 149.3±1.4 | 0.08 | 174.7±1.7 | 0.08 | 104.7±1.6 | 0.08 |
| COPA ($\beta = 8$) | 109.4±3.6 | 0.04 | 130.6±4.0 | 0.04 | 80.6±2.0 | 0.04 |

Table 1: Generalization performance on unseen environments with more agents and *dynamic team composition*. Results are computed from 5 models trained with 5 different seeds. Communication frequency is compared to communicating with all agents at every step.

**Zero-shot Generalization** We apply the learned model with $T = 4$ to the 3 testing environments. Results are provided in Table 4.1. The communication frequency is calculated according to the fully centralized setting. For instance, when $T = 4$ and $\beta = 0$, it results in an average $25\%$ centralization frequency. As we increase $\beta$ to suppress the distribution of strategies, we see that the performance shows no significant drop till $13\%$ centralization frequency. Moreover, we apply the same model to 3

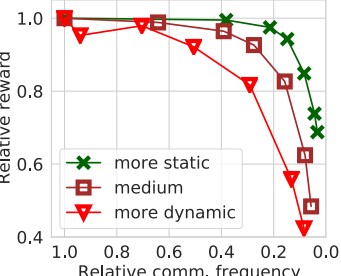

Figure 5: The varying sensitivity to communication frequency.

environments that are dynamic to different extents. In the more static environment, resources are always spawned at the same locations. In medium environment, resources are spawned randomly but there is no invader. The more dynamic environment is the 3rd environment in Table 4.1 where the team is dynamic in composition and there exists the invader. Result is summarized in Figure 5. Here, the x-axis is normalized according to the communication frequency when $\beta = 0$, and the y-axis is normalized by the corresponding performance. As expected, as the environment becomes more dynamic, low communication frequency more severely downgrades the performance.

## 4.2 RESCUE GAME

Search-and-rescue is a natural application of multi-agent systems. In this section we further apply COPA to a rescue game. In particular, we consider a $10 \times 10$ grid-world, where each grid contains a building. At any time step, each building is subject to catch a fire. When a building $b$ is on fire, it has an emergency level $c^b \sim \mathcal{U}(0, 1)$. Within the world, at most 10 buildings will be on fire at the same time. Fortunately we have $n$ ($n$ is a random number from 2 to 8) robots who are the surveillance firefighters. Each robot $a$ has a skill-level $c^a \in [0.2, 1.0]$. A robot has 5 actions, moving up/down/left/right and put out the fire. If $a$ is at a building on fire and chooses to put out the fire, the emergency level will be reduced to $c^b \leftarrow \max(c^b - c^a, 0)$. At each time step $t$, the overall-emergency is given by $c_t^B = \sum_b (c^b)^2$ since we want to penalize the existence of more emergent fire. The reward is defined as $r_t = c_{t-1}^B - c_t^B$, the amount of emergence level the team reduces. During training, we sample $n$ from $3 - 5$ and these robots are spawned randomly across the world. Each agent's skill-level is sampled from $[0.2, 0.5, 1.0]$. Then a random number of $3 - 6$ buildings will catch a fire. During testing, we enlarge $n$ to $2 - 8$ agents and sample up to 10 buildings on fire. We summarize the result in the Table 4.2. Interestingly, we find that A-QMIX with full observation

| | Random | Greedy | A-QMIX | A-QMIX (full) | COPA (w/o $\mathcal{L}_{\text{var}}$) | COPA (1) | COPA (0.5) | COPA (0.15) |
|---|---|---|---|---|---|---|---|---|
| Epi. Reward | 1.4 | 7.0 | 5.4±0.5 | 1.6±0.5 | 9.0±0.6 | 10.7±0.6 | 11.1±0.8 | 8.9±0.5 |

Table 2: Average episodic reward over the same 500 Rescue games. Results are averaged over the same algorithm trained with 3 different seeds. For COPA $(x)$, $x$ denotes the communication frequency. Greedy algorithm matches the $k$-th skillful agent for the $k$-th emergent building.

failed to learn. We conjecture this is because the team has too much information to process during training and therefore it is hard to search for a good policy. COPA consistently outperforms all baselines even with a communication frequency as low as 0.15.

## 5 RELATED WORKS

In this section we briefly go over some related works in cooperative multi-agent reinforcement learning and hierarchical reinforcement learning.

**Centralized Training with Decentralized Execution**  Centralized training with decentralized execution (CTDE) assumes agents execute independently but uses the global information for training. A branch of methods investigates factorizable $Q$ functions (Sunehag et al., 2017; Rashid et al., 2018; Mahajan et al., 2019; Son et al., 2019) where the team $Q$ is decomposed into individual utility functions. Some other methods adopt actor-critic method where only the critic is centralized (Foerster et al., 2017; Lowe et al., 2017). However, most deep CTDE methods by structure require fixed-size teams and are often applied on homogeneous teams.

**Methods for Dynamic Compositions**  Several recent works pay attention to transfer learning and curriculum learning in MARL problems where the learned policy is is a warm start for new tasks (Carion et al., 2019; Shu & Tian, 2018; Agarwal et al., 2019; Wang et al., 2020b; Long et al., 2020). These works focus on curriculum learning and mostly consider homogeneous agents. Hence the team strategy is relatively consistent. Iqbal et al. (2020) first adopt the multi-head attention mechanism for dealing with a varying size heterogeneous team. But the heterogeneity comes from a small finite set of agent types (usually 2 to 3). Additionally, the method is fully decentralized and therefore less adaptive to frequent change in team composition.

**Ad Hoc Teamwork and Networked Agents**  Ad hoc teamwork studies the problem of quick adaptation to unknown teams (Genter et al., 2011; Barrett & Stone, 2012). However, ad hoc teamwork

focuses on the single ad hoc agent and often assumes no control over the teammates and therefore is essentially a single-agent problem. Decentralized networked agents assume information can propagate among agents and their neighbors (Kar et al., 2013; Macua et al., 2014; Suttle et al., 2019; Zhang et al., 2018). However, research in networked agents still mainly focus on homogeneous fixed-size teams. Although it is possible to extend the idea for the dynamic team composition problem, we leave it as a future work.

**Hierarchical Reinforcement Learning** The main focus of hierarchical RL/MARL is to decompose the task into hierarchies: a meta-controller selects either a temporal abstracted action (Bacon et al., 2017), called an option, or a goal state (Vezhnevets et al., 2017) for the base agents. Then the base agents shift their purposes to finish the assigned option or reach the goal. Therefore usually the base agents have different learning objective from the meta-controller. Recent deep MARL methods also demonstrate role emergence (Wang et al., 2020a) or skill emergence (Yang et al., 2019). But the inferred role/skill is only conditioned on the individual trajectory. The coach in our method uses global information to determine the strategies for the base agents. To our best knowledge, we are the first to consider applying such hierarchy for teams with varying number of heterogeneous agents.

## 6    CONCLUSION

We investigated a new setting of multi-agent reinforcement learning problems, where both the team size and members' capabilities are subject to change. To this end, we proposed a coach-player framework where the coach coordinates with global view but players execute independently with local views and the coach's strategy. We developed a variational objective to regularize the learning and introduces an intuitive method to suppress unnecessary distribution of strategies. The experiment results across multiple unseen scenarios on the Resource Collection task demonstrate the effectiveness of the coach agent. The zero-shot generalization ability of our method shows a promising direction to real-world ad hoc multi-agent coordination.

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

## A APPENDIX

### PROOF OF THEOREM 1

Here we expand the assumptions from Theorem 1 and provide the proof for it. The two assumptions are:

**Assumption 1.** *Denote the learned team action-value function as $Q^{tot}$, the learned coach strategy encoder as $f$ and the true optimal action-value function as $Q_*^{tot}$. We assume for any $\boldsymbol{\tau}_t, \boldsymbol{u}_t, \boldsymbol{s}_t, \boldsymbol{s}_{\hat{t}}, \boldsymbol{c}$,*

$$||Q^{tot}(\boldsymbol{\tau}_t, \boldsymbol{u}_t, f(\boldsymbol{s}_{\hat{t}}); \boldsymbol{c}) - Q_*^{tot}(\boldsymbol{s}_t, \boldsymbol{u}_t; \boldsymbol{c})||_2 \leq \kappa. \tag{7}$$

**Assumption 2.** *Denote the learned individual action-value function as $\{Q^{a_i}\}_{i=1}^{n_a}$, and the particular individual action-value at a state $\boldsymbol{s}$ with action $\boldsymbol{u}$ as $\{q^{a_i} = Q^{a_i}(s^{a_i}, u^{a_i})\}_{i=1}^{n_a}$. Then we assume unilaterally varying any $q^{a_i}$ to $q'$, i.e. all other $\boldsymbol{q}^{-a_i}$ remain the same, will not cause dramatic change of $Q^{tot}$ if $q'$ stays closely to $q^{a_i}$:*

$$\left|Q^{tot}(\boldsymbol{q}^{-a_i}, q^{a_i}) - Q^{tot}(\boldsymbol{q}^{-a_i}, q')\right| \leq \eta_1 |q^{a_i} - q'| \tag{8}$$

*and for any agent $a$ and $\forall c^a, \tau_t^a, u_t^a, z_1^a, z_2^a$ with proper dimensions,*

$$\left|Q^a(\tau_t^a, u_t^a | z_1^a; c^a) - Q^a(\tau_t^a, u_t^a | z_2^a; c^a)\right| \leq \eta_2 ||z_1^a - z_2^a||_2. \tag{9}$$

In other words, assumption 1 assumes the learned $Q^{\text{tot}}$ approximates the true optimal $Q_*^{\text{tot}}$ well *combined with* the learned coach strategy function $f$,[6] and assumption 2 assumes the learned team action-value $Q^{\text{tot}}$ has bounded Lipschitz constant. Next we provide the proof for Theorem 1.

---

[6]Note here we only assume $Q^{\text{tot}}$ is accurate around the predicted strategy by $f$, not for any strategy.

*Proof.* From assumption 2, it is easy to check that if $||\tilde{z}_t^a - z_{\hat{t}}^a||_2 \leq \beta$ for all $a$, then $|Q^{\text{tot}}(\boldsymbol{\tau}_t, \boldsymbol{u}_t|\tilde{\boldsymbol{z}}_t, \boldsymbol{c}) - Q^{\text{tot}}(\boldsymbol{\tau}_t, \boldsymbol{u}_t|\boldsymbol{z}_{\hat{t}}, \boldsymbol{c})| \leq n_a \eta_1 \eta_2 \beta$. For notation convenience, we ignore the superscript of tot and the condition on $\boldsymbol{c}$. For a state $\boldsymbol{s}$, denote the action the learned policy take as $\boldsymbol{u}^\dagger$, $\boldsymbol{u}^\dagger = \arg\max_{\boldsymbol{u}} Q(\boldsymbol{\tau}, \boldsymbol{u})$. Similarly we can define $\boldsymbol{u}^*$ as the action the optimal $Q_*$ takes and $\tilde{\boldsymbol{u}}$ that $\tilde{Q}$ takes. From assumption 1, we know that

$$Q_*(\boldsymbol{s}, \boldsymbol{u}^\dagger) \geq Q(\boldsymbol{\tau}, \boldsymbol{u}^\dagger) - \kappa \geq Q(\boldsymbol{\tau}, \boldsymbol{u}^*) - \kappa \geq Q_*(\boldsymbol{s}, \boldsymbol{u}^*) - 2\kappa. \tag{10}$$

Therefore taking $\boldsymbol{u}^\dagger$ will result in at most $2\kappa$ performance drop at this single step. Similarly, denote $\epsilon_0 = n_a \eta_1 \eta_2 \beta$, then

$$Q(\boldsymbol{\tau}, \tilde{\boldsymbol{u}}) \geq \tilde{Q}(\boldsymbol{\tau}, \tilde{\boldsymbol{u}}) - \epsilon_0 \geq \tilde{Q}(\boldsymbol{\tau}, \boldsymbol{u}^\dagger) - \epsilon_0 \geq Q(\boldsymbol{\tau}, \boldsymbol{u}^\dagger) - 2\epsilon_0. \tag{11}$$

Hence $Q_*(\boldsymbol{s}, \tilde{\boldsymbol{u}}) \geq Q_*(\boldsymbol{s}, \boldsymbol{u}^*) - 2(\epsilon_0 + \kappa)$. Note that this means taking the action $\tilde{\boldsymbol{u}}$ in the place of $\boldsymbol{u}^*$ at state $\boldsymbol{s}$ will result in at most $2(\epsilon_0 + \kappa)$ performance drop. This conclusion generalizes to any step $t$. Therefore, if at each single step the performance is bounded within $2(\epsilon_0 + \kappa)$, then overall the performance is within $2(\epsilon_0 + \kappa)/(1 - \gamma)$. $\qquad\square$

## NETWORK ARCHITECTURE

For all experiments, we use the same network architecture where all intermediate hidden layer have 128 dimensions. Note that this is possible since the only difference is the number of entities, which does not influence our architecture when adopting an attention model. The architecture details follow exactly as in Appendix A of (Iqbal et al., 2020).

## TRAINING DETAILS

To train the model, we set the max total number of steps to 5 million. Then we use the exponentially decayed $\epsilon$-greedy algorithm as our exploration policy, starting from $\epsilon_0 = 1.0$ to $\epsilon_n = 0.05$. We parallel the environment with 8 threads for training. Details on hyper-parameters are available in Section A.

## HYPER PARAMETERS

For all experiments, we use the same set of hyper-parameters. We provide them in the following table:

| Name | Description | Value |
| --- | --- | --- |
| $|\mathcal{D}|$ | replay buffer size | 100000 |
| $n_{\text{head}}$ | number of heads in multi-head attention | 4 |
| $n_{\text{thread}}$ | number of parallel threads for running the environment | 8 |
| $dh$ | the hidden dimension of all modules | 128 |
| $\gamma$ | the discount factor | 0.99 |
| $lr$ | learning rate | 0.0003 |
| | optimizer | RMSprop |
| $\alpha$ | $\alpha$ value in RMSprop | 0.99 |
| $\epsilon$ | $\epsilon$ value in RMSprop | 0.00001 |
| $n_{\text{batch}}$ | batch size | 256 |
| grad clip | clipping value of gradient | 10 |
| target update frequency | how frequent do we update the target network | 200 updates |
| $\lambda_1$ | $\lambda_1$ in variational objective | 0.001 |
| $\lambda_2$ | $\lambda_2$ in variational objective | 0.0001 |

Table 3: Hyper-parameters in our experiments.

