# OpenReview forum: "A Coach-Player Framework for Dynamic Team Composition"
_ICLR.cc/2021/Conference — Reject_

### Official Review · AnonReviewer3 · 2020-10-23
**ICLR 2021 Conference Paper2657 AnonReviewer3**

**Rating:** 7
**Confidence:** 2

**Review:**

Summary:

This paper studies the dynamic multi-agent team coordination problem, in which the optimal team strategy may change over time as the environment and the team members vary. The authors propose a coach-player framework in which only the coach has full information about the game while the players only have their own local information, separately. In this framework, the coach computes the optimal game plan and sends the plan to the players periodically, and the players play according to the received plan from the coach and their local information. Empirical studies demonstrate the effectiveness of the new framework.

Comments:

This paper is generally well-written and clear. The idea of having a coach to compute the optimal game plan to coordinate the players seems interesting and non-trivial. In the empirical study, COPA with variational objective outperforms the state-of-art benchmark. The authors also have an interesting observation that always communicating the optimal global game plan to the players is not always the optimal strategy.

1. The reviewer is not familiar with A-QMIX. Is it correct that the main difference between A-QMIX (full) and COPA is that in A-QMIX, even though the players have access to the full information, they still compute for their own game plans instead of coming up with a joint optimal game plan for the team?

2. As the authors claim that the coach is more useful when it can make the agents behavior smooth/consistent over time, the reviewer is wondering whether it would be helpful to communicate a smoothed strategy update to the players. More precisely, the current way is to set a threshold on the distance between z_t and z_old to limit the communication frequency. For example, if the coach always communicates, whether it would be helpful if the coach sends a convex combination between z_t and z_old (similar to ideas of online learning) to smooth the strategy update?

---

> ### Author Response · Authors · 2020-11-16
> **Author Response to R3**
>
> We thank the reviewer for these comments. Please also refer to the response to common questions above for more information. Here we address the reviewer's concerns in more details.
>
> **Q1**: The reviewer is not familiar with A-QMIX. Is it correct that the main difference between A-QMIX (full) and COPA is that in A-QMIX, even though the players have access to the full information, they still compute for their own game plans instead of coming up with a joint optimal game plan for the team?
>
> **A1**: Yes, in A-QMIX, each agent will compute their own plans without a global centralized agreement in contrast to COPA. But in principle it is also likely that during training, each agent learns to perform specific behaviors for coordination. This is how A-QMIX and other CTDE methods learn to collaborate with individual execution models. However, in our setup, since we consider teams with more dynamic composition, the coordination becomes much harder to learn without any global coordination (the coach in our case). Moreover, as indicated in both experiments, A-QMIX with full observation is essentially a fully centralized method where all agents know all the information. The comparison against A-QMIX with full observation indicates that purely feeding the agent with more information will not help the learning, though in principle this should be the strongest method.
>
> **Q2**: As the authors claim that the coach is more useful when it can make the agents behavior smooth/consistent over time, the reviewer is wondering whether it would be helpful to communicate a smoothed strategy update to the players. More precisely, the current way is to set a threshold on the distance between $z_t$ and $z_\text{old}$ to limit the communication frequency. For example, if the coach always communicates, whether it would be helpful if the coach sends a convex combination between $z_t$ and $z_\text{old}$ (similar to ideas of online learning) to smooth the strategy update?
>
> **A2**: We thank the reviewer for this interesting suggestion. We think it is possible that more smooth communication will benefit the learning and execution. But on the other hand, it is likely that upon a specific environment change, the team needs to sharply modify its strategy (for example with loss of a few members, or the appearance of the invader in our experiment). In that case, we do want the coach to send a drastically different strategy for the particular agent.

---

### Official Review · AnonReviewer2 · 2020-10-25
**Marginally above acceptance**

**Rating:** 6
**Confidence:** 3

**Review:**

Summary:
The authors propose a coach-player framework for dynamic team composition of dynamic and heterogeneous agents based on deep Q learning with an attention mechanism and a variational objective to regularize the learning. The authors design an adaptive communication strategy to minimize communication from the coach to the agents. Using a resource-collection task in multi-agent particle environments, the authors evaluated zero-shot generalization for new team compositions at test time. Results show comparable or even better performance against methods where players have full observation but no coach. Interestingly, there is almost no performance degradation even when the coach communicates as little as 13% of the time with the players.

Reasons for score:
Although the motivation, novelty compared with the related work, and a superiority of the proposed method in an experimental result were clear, there were unclear points in the background and methods (without sharing codes) and the authors performed only a simple experiment. I think the idea is interesting and contributed to this community, but for the above reasons it is difficult to provide a higher rating.


Pros:
1. The motivation and novelty compared with the related work were clear.
2. The contribution in this paper is to propose a coach-player framework for dynamic team composition of dynamic and heterogeneous agents based on deep Q learning with an attention mechanism and a variational objective to regularize the learning.
3. In a resource-collection task in multi-agent particle environments, the proposed method clearly demonstrated the effectiveness of having a coach in dynamic teams.

Cons:
1. There were many unclear points in Sections 2 and 3 (methodology, see below)
2. The authors performed only one experiment in a resource-collection task.  It seems to be a relatively simple setting for me, but the investigation in another experiment will help us understand the effectiveness of the proposed method from more general perspectives.

Other comments:

Regarding deep recurrent Q-learning (Zhu et al., 2017), was u_0^a in "tau^a = (o_0^a, u_0^a,... o_t^a) correct?  If correct, more formal expression and additional explanation may be required.

Did u’ in eqs. (1) and (2) mean u_{t+1}? It may be confusing.

Was 2.2 VALUE FUNCTION action-value function or Q function?

“Speciﬁcally, the input o is represented by two matrices…” Was the input tau rather than o in 2.2? What is the input?

I did not understand the first equation of eq. (4). It seems to be different from the definition of the mutual information that I know, and the idea of the referenced papers (Rakelly et al., 2019; Wang et al., 2020a). A careful introduction is required.

======after rebuttal=========

Thank you for answering my questions.  I understood these points. The authors added a new simple experiment and a code, whereas the manuscript at the current stage can improve the clarity. All things considered, I increased the rating.

---

> ### Author Response · Authors · 2020-11-16
> **Author Response to R2**
>
> We thank the reviewer for these comments. Please also refer to the response to common questions above for more information. Here we address the reviewer's concerns in more details.
>
>
> **Q1**: "Regarding deep recurrent Q-learning (Zhu et al., 2017), was $u_0^a$ in $\tau^a = (o_0^a, u_0^a,... o_t^a)$ correct? If correct, more formal expression and additional explanation may be required."
>
> **A1**: Yes, it is correct. In practice, at time step $t$, the RNN will take inputs of $(o_t, u_{t-1})$, the current observation and previous action. Therefore, for time step $t=0$, we feed $(o_0, \vec{0})$ to the network, where $\vec{0}$ is the all-zero vector. We have clarified this point in the paper.
>
> **Q2**: "Did $u’$ in eqs. (1) and (3) mean $u_{t+1}$? It may be confusing.
>
> **A2**: In both eqs. (1) and (3), $u’$ here is a placeholder action that runs over all possible actions at time $t+1$. $u_{t+1}$ ideally should be the argmax among all such $u’$.
>
> **Q3**: Was 2.2 value function the action-value function or Q function?
>
> **A3**: Yes, value function here refers to the action-value function, a.k.a the Q function. Here we use the term “value function factorization” as it is used in the title of the original QMIX paper [1].
>
> **Q4**: “Speciﬁcally, the input $o$ is represented by two matrices…” Was the input $\tau$ rather than $o$ in 2.2? What is the input?
>
> **A4**: Here, the input $o$ is part of the *per-step* input fed into the RNN (the other part is the previous action $u_{t-1}$). On the other hand, since the RNN summarizes the history (all prior observation $o$ and actions $u$) up to current time $t$, the decision is conditioned on the past history $\tau$, i.e. $\tau = (o_0, u_0, \dots, u_{t-1}, o_t)$.
>
> **Q5**: I did not understand the first equation of eq. (4). It seems to be different from the definition of the mutual information that I know, and the idea of the referenced papers (Rakelly et al., 2019; Wang et al., 2020a). A careful introduction is required.
>
> **A5**: Thanks for asking. We reference the two works here just to indicate they motivate us to consider adding additional variational objectives to regularize the learning. Regarding the exact derivation, we refer the reviewer to equation (2) from [2], i.e. $ I(X | Y) \geq E_{p(x,y)}[\log q(x | y)]  + H(X).$ In our case, the strategy $z_t$ is the $X$ and ($\zeta$ and $s_t$) are the $Y$.
>
> **Reference**
> [1] Rashid, Tabish, et al. "QMIX: Monotonic value function factorisation for deep multi-agent reinforcement learning." arXiv preprint arXiv:1803.11485 (2018).
>
> [2] Poole, Ben, et al. "On variational bounds of mutual information." arXiv preprint arXiv:1905.06922 (2019).

---

### Official Review · AnonReviewer1 · 2020-10-27
**In its current form, I find the paper’s line of argumentation, as well as the experiments, to detached from relevant prior work wrt. learned communication within/for MARL. Further, I believe, the paper would benefit from disentangling the communication idea with the variational objective, which is more in line with hierarchical approaches (which is accounted for in the related work section).**

**Rating:** 4
**Confidence:** 3

**Review:**

Summary:
The paper proposes to extend the centralized training, decentralized execution paradigm for cooperative multi-agent RL with a coach-player framework. The proposed framework would allow the coach to have access to the full observation, but only allow for limited communication of a continuous strategy vector to the other agents / players. It is argued, that the framework helps with dynamic composition of multi-agent teams, which encompasses variable numbers of agents, as well as heterogeneous agents.
A variational objective is introduced, to force the player trajectories to be coherent with the strategy assigned by the coach. A concrete algorithm is proposed, that builds strongly upon [1]. Experimental evidence shows, that the coach-player framework with variational objective outperforms a baseline from this paper (referred to as A-QMIX) even if the latter is granted full observability for its agents.

Strong:
Considering the addition of a coach agent with constraint communication makes intuitively sense and there very well might be scenarios, where this addition is useful to solve actual tasks. The paper provides experimental evidence, that structured communication about the coordination of decentralized agents is beneficial, even if the communication frequency is enforced to be sparse. The variational objective seems to be especially helpful. A proof is included, which shows that sparse communication results in only small loss compared to more frequent communication if the representation of the team strategy is relatively stable (wrt. changes in its L2 norm). The paper includes results which suggest good generalization performance of the proposed algorithm wrt. unseen team composition.

Weak:
Mainly, I consider the papers experimental evaluation to be not very convincing. This might very well be due to the paper proposing a specific type of communication (with global knowledge), while disregarding to argue about different types of communication in MARL and only comparing to an approach that does not explicitly involve communication. This seems especially strange, since the multi-agent particle environment upon which the experiments build [2], was explicitly contrived to analyze communication (grounded in RL scenarios). The comparison of the proposed algorithm (referred to as COPA) to the baseline (A-QMIX) alone does not seem very insightful. Both show about the same performance if A-QMIX agents are provided with full observations and COPA is ablated wrt. its variational objective. That COPA’s performance for this task seems dependent on the variational objective (which is not further discussed), to me is an indication that important issues are not addressed in the paper / the experiments. Given its focus, I further argue that the paper lacks related literature with respect to learned communication in MARL [e.g. compare with listed literature in a current survey, 3].

Conclusion:
In its current form, I find the paper’s line of argumentation, as well as the experiments, to detached from relevant prior work wrt. learned communication within/for MARL. Further, I believe, the paper would benefit from disentangling the communication idea with the variational objective, which is more in line with hierarchical approaches (which is accounted for in the related work section).

 Other comments:
The AI-QMIX reference [1] changed significantly after the paper submission. An update of the reviewed paper with respect to this should be considered. Also, it is easy to get lost in the very involved notation. (Wrt. this: Are the subscripts to z consistent in Section 3.4; especially in the paragraph leading to Theorem 1?)

[1] Iqbal, Shariq; Witt, Christian A. Schroeder de; Peng, Bei; Böhmer, Wendelin; Whiteson, Shimon; Sha, Fei (2020): Randomized Entity-wise Factorization for Multi-Agent Reinforcement Learning. In: arXiv preprint arXiv:2006.04222.
[2] Mordatch, Igor; Abbeel, Pieter (2018): Emergence of Grounded Compositional Language in Multi-Agent Populations. In: AAAI Conference on Artificial Intelligence.
[3] Hernandez-Leal, Pablo; Kartal, Bilal; Taylor, Matthew E. (2019): A survey and critique of multiagent deep reinforcement learning. In: Autonomous Agents and Multi-Agent Systems 33 (6), S. 750–797

---

> ### Author Response · Authors · 2020-11-16
> **Author Response to R1**
>
> We thank the reviewer for these comments. Please also refer to the response to common questions above for more information. Here we address the reviewer's concerns in more details.
>
> As mentioned in the response to common concerns, we find that learning to communicate in ad-hoc heterogeneous teams is less studied in the literature compared to that in common homogeneous teams. While some existing methods work on a known communication protocol, it requires human efforts to create such protocols. On the other hand, A-QMIX is one of the first to coordinate a set of agents. A-QMIX with full observation, in particular, can both handle the change in team composition and coordinate the team globally. Therefore, we believe the comparison against A-QMIX with full observation is fair. The results demonstrate that purely granting agents with more information will not help (or even hurt) the learning.
>
> **Q1**: "That COPA’s performance for this task seems dependent on the variational objective (which is not further discussed), to me is an indication that important issues are not addressed in the paper / the experiments"
>
> **A1**: we want to emphasize that A-QMIX with full observation means that all agents have access to the global information at every time step. By contrast, in COPA, every agent gets the strategy from the coach at most once every $T$-step. Therefore, A-QMIX with full observation is supposed to be an upper-bound on the performance.
>
> **Q2**: "Given its focus, I further argue that the paper lacks related literature with respect to learned communication in MARL [e.g. compare with listed literature in a current survey, 3]."
>
> **A2**: Thanks for you advice and we will include a more thorough related works section that also address the learned communication in MARL. But as mentioned in the response to common concerns, most works addressing the learned communications in MARL study the homogeneous fixed-size team. In our setup where team is subject to change, having the coach agent is a more effective way of communication. We believe our work is a reasonable first step to efficient communication in ad-hoc teamwork.
>
> Regarding the update on the AI-QMIX reference, we have checked the updated paper. We find that the update is mainly on the experiment part while the main algorithm remains the same. Therefore all of our comparisons against their method remain valid.

---

### Official Review · AnonReviewer4 · 2020-10-28
**An approach to improving the effectiveness of decentralised cooperative learning with limited novelty and validation**

**Rating:** 5
**Confidence:** 3

**Review:**

This paper addresses the important problem of being able to deal with heterogeneous teams of agents (that might change over time) in cooperative multiagent reinforcement learning. In the strictly cooperative setting with identical problem representations among agents, this essentially boils down to a problem of decentralized learning and control, where, effectively, the improvements demonstrated by the paper over existing papers boil down to effectively reducing the ways in which non-local information needs to be communicated between a global observer (coach) and the individual learning (agents). The specific novelty of the approach is to introduce a variational objective to stablise training, and introduce a heuristic for reducing the communication frequency.

As is the case with many similar papers in deep reinforcement learning, much of the paper's approach is premised on pre-training high-dimensional function approximators with an enormous amount of data to learn an optimal strategy offline that can then be applied in sequential games, in this case a reasonably complex (yet toy) scenario. Only a small part of the paper focuses on the technical innovations while the bulk of it describes the application of DNN techniques to the specific coach-player mechanism in a specific game. In my view, this does not provide enough evidence to be able to assess the general applicability of the approach, in particular because it is only evaluated in a single domain and much of the reported performance could be attributed to fine-tuning of domain-specific parameters.

The paper is generally well-written and relatively easy to follow, though many of the design decisions are not very well explained, and there is no explicit problem formulation regarding what optimal communication strategies would look like, which would help evaluate the advances reported in the paper with respect to the overall fundamental problem. As is the case with many similar papers, while the existing work is surveyed, I find comparisons to other approaches somewhat unfair, as many of them propose actual algorithms that deal with the problem of coordination at runtime, rather than creating than making (as is proposed here) certain small modifications to immensely data-hungry function approximation methods that learn an effective coordination policy from data, and offer little insight on what it actually looks like that advances our understanding of the problem to build better future AI systems.

The language is not perfect in some places, but could be easily polished with additional proofreading.

---

> ### Author Response · Authors · 2020-11-16
> **Author Response to R4**
>
> We thank the reviewer for these comments. Please also refer to the response to common questions above for more information. Here we address the reviewer's concerns in more details.
>
> Regarding the **technical innovations**, we think all the key contributions (coach-player hierarchy, variational objective, and the communication criterion) focus on making efficient usage of the global information, which is essential for teams with dynamic compositions (as discussed in Section 3.1). By comparing against A-QMIX with full observation, which is essentially a centralized learning method, we see that simply feeding the global information for each agent does not help (or can even hurt) the learning. While our contributions are closely related to deep learning techniques, we want to convince the reviewer that they are all based on close observations of the dynamic team composition problem itself.
>
> Regarding the **stability of the performance** of our method, upon the two experiments we have conducted, the performance is consistent without careful fine-tuning. In fact, we list the hyper-parameters we use in the Appendix, most of which are picked without any tuning at all. We have also included the code for reproducing our results.
>
> Regarding **prior works that do online coordination**, we believe most of such works assume a known distribution to at least of the underlying tasks, roles, and communication protocols [1,2].  In fact, we believe for ad-hoc team coordination, the agents more or less need some prior knowledge about their teammates. These knowledge can be either in the form of pre-specified roles/rules (even pre-trained low-level controls [3]) or pre-trained team behavior (as in our work). One benefit of pre-training the team behavior is that there is no need to design specific roles/rules manually, this is particularly useful in applications where manual design of a collaboration rule/role assignment is extremely hard.
>
> **Reference**
> [1] Barrett, Samuel, et al. "Communicating with Unknown Teammates." ECAI. 2014.
>
> [2] Genter, Katie Long, Noa Agmon, and Peter Stone. "Role-Based Ad Hoc Teamwork." Plan, Activity, and Intent Recognition. 2011.
>
> [3] Carion, Nicolas, et al. "A structured prediction approach for generalization in cooperative multi-agent reinforcement learning." Advances in Neural Information Processing Systems. 2019.

---

### Author Response · Authors · 2020-11-16
**Author Response to Common Questions**

We thank all reviewers for their insightful comments. We address the common concerns here and will respond to individual reviewers to address their specific concerns separately.

**Key points motivating the design choices**
1. We study how to coordinate teams of dynamic composition (both the team size and agents' characteristics are subject to change);
2. We emphasize it is important to have ``some type of global communication for efficient strategy adaptation under the problem of dynamic team composition;
3. Therefore, the main contributions of the paper all focus on how to efficiently make use of the global information for coordination. Specifically, a) We decompose the learning into hierarchies. Therefore the agents learn local skills from local observation while the coach learns the team-level strategy for coordination with the full information; b) The introduced variational objective essentially limits the policy's search space due to the inductive bias. Therefore, with this regularization, our method can achieve faster learning or even better final performance; c) Finally we introduce the criterion for more efficient communication for more practical application of our method.

**Experiment** : we have conducted an additional experiment on Rescue games (Section 4.2), where a team of firefighters with different skill-levels collaborate to put out fires in a 10x10 grid-world. We provided the result here. We have also included it in the updated paper.

|                    |Greedy$~~$|A-QMIX$~~$|A-QMIX (full)$~~$|COPA(no $\mathcal{L}_{\text{var}}$)$~~$|COPA (1)$~~$|COPA (0.5)$~~$|COPA (0.15)$~~$|
|:-----:|:-----:|:-----:|:-----:|:-----:|:-----:|:-----:|:-----:|
|Reward |7.0| 5.4$\pm$0.5|1.6$\pm$0.5|9.0$\pm$0.6|10.7$\pm$0.6|11.1$\pm$0.8|8.9$\pm$0.5|

Here, COPA ($x$) denotes COPA with communication frequency $x$. Interestingly, A-QMIX with full observation almost failed to learn, which further suggests it is beneficial to decompose the learning into hierarchies. Note that COPA consistently outperforms the baseline methods even with a communication frequency as low as 0.15.

**Reproducibility**: We submit our code in supplementary materials for reproducing our experiment results.

**Comparison to previous work**: We want to point out some key differences from previous literature. While prior works in networked agents have studied learned communication for coordination, it is mostly applied to homogeneous teams of fixed size. For ad-hoc teamwork, prior research mostly focus on the single ad-hoc agent that learns to adapt to the teammates (in contrast to modifying the entire team strategy). When communication is used in ad-hoc teams, the communication protocol is usually assumed to be pre-defined [1]. Or the communication protocol is pre-defined for all agents than the ad-hoc agent [2]. A more recent work that we are aware of which studies the communication in ad-hoc teams is [3]. But it is also based on the assumption that agents have the common knowledge to understand the queries from others. In general, we believe learned communication in ad-hoc teams is less investigated than that in homogeneous teams. However, we do think that a fully online ad-hoc communication scheme is an interesting research direction for future work. On the other hand, as discussed in Section 3.1, local communication still takes time to propagate through the entire team, which in general is less efficient than having a centralized agent to which all agents communicate.

**Reference**
[1] Barrett, Samuel, et al. "Communicating with Unknown Teammates." ECAI. 2014.

[2] Grizou, Jonathan, et al. "Collaboration in Ad hoc teamwork: ambiguous tasks, roles, and communication." AAMAS Adaptive Learning Agents (ALA) Workshop. 2016.

[3] Mirsky, Reuth and Macke, William and Wang, Andy and Yedidsion, Harel and Stone, Peter. “A Penny for Your Thoughts: The Value of Communication in Ad Hoc Teamwork.” (IJCAI-20)

---

### Decision · Program_Chairs · 2021-01-07
**Final Decision**

**Decision:**

Reject

**Comment:**

This paper proposes an approach for coordinating teams with dynamic composition consisting of an attention mechanism, regularization and communication. The clarity of the paper is currently low seemingly due to the conflated message of the multiple parts of the framework. Improvements to the text via the suggested edits of all reviewers should be a relatively quick fix, but the clearer placement of this piece within the wider literature may require additional experiments to compare against so would be a larger change.

The reviewers did continue to discuss the paper after the end of the open discussion period with the authors and appreciated the additional experiments performed. In the absence of supporting theory, empirical results in a second domain significantly improve the evidence that the method may be more generally applicable. However, the new experiments raised new questions (included in the reviewers later replies) indicating more experiments in the second domain are needed which would require further peer review.

I hope the authors will take the constructive feedback provided here as intended; to improve the paper, submit the work again at a later stage when the second experimental domain is sufficiently explored to support the proposed framework and in doing so maximise its potential for impact.